# Transcription Factor TaMYB30 Activates Wheat Wax Biosynthesis

**DOI:** 10.3390/ijms241210235

**Published:** 2023-06-16

**Authors:** Lang Liu, Haoyu Li, Xiaoyu Wang, Cheng Chang

**Affiliations:** College of Life Sciences, Qingdao University, Qingdao 266071, China

**Keywords:** wheat, wax biosynthesis, MYB30, KCS1, ECR

## Abstract

The waxy cuticle covers a plant’s aerial surface and contributes to environmental adaptation in land plants. Although past decades have seen great advances in understanding wax biosynthesis in model plants, the mechanisms underlying wax biosynthesis in crop plants such as bread wheat remain to be elucidated. In this study, wheat MYB transcription factor TaMYB30 was identified as a transcriptional activator positively regulating wheat wax biosynthesis. The knockdown of *TaMYB30* expression using virus-induced gene silencing led to attenuated wax accumulation, increased water loss rates, and enhanced chlorophyll leaching. Furthermore, *TaKCS1* and *TaECR* were isolated as essential components of wax biosynthetic machinery in bread wheat. In addition, silencing *TaKCS1* and *TaECR* resulted in compromised wax biosynthesis and potentiated cuticle permeability. Importantly, we showed that TaMYB30 could directly bind to the promoter regions of *TaKCS1* and *TaECR* genes by recognizing the *MBS* and *Motif 1* cis-elements, and activate their expressions. These results collectively demonstrated that TaMYB30 positively regulates wheat wax biosynthesis presumably via the transcriptional activation of *TaKCS1* and *TaECR*.

## 1. Introduction

The lipophilic cuticle covers the above-ground portions of plants and represents a true interphase between plant aerial organs and surrounding environments [1,2,3]. The cuticle is mainly composed of cutin matrices filled and covered with wax mixtures [1,2,3]. Through preventing organ fusion and promoting root formation, the cuticle tightly regulates plant development [4,5,6]. In addition, the waxy cuticle greatly contributes to the plant’s adaption to environmental challenges associated with terrestrialization such as water deficit (drought), high salinity, extreme temperatures (cold and heat), ultraviolet (UV) radiation, and attacks from pathogens and pests (P&Ps) [7,8,9,10,11]. Increasing studies suggested that cuticle-associated traits have been selected and/or manipulated for genetic improvement in important crop plants such as hexaploid bread wheat (*Triticum aestivum*) [12,13].

Plant cuticular wax mixtures consist of very long-chain (VLC, >C20) fatty acids and their derivatives such as alcohols, alkanes, aldehydes, ketones, and esters [1,2,3]. As summarized by prior reviews, cuticle wax biosynthesis mainly occurs in the endoplasmic reticulum (ER) of plant epidermal cells [1,2,3]. Briefly, C16 and C18 fatty acids trafficked from the plastid are first esterified with coenzyme A (CoA) by long-chain acyl-coenzyme A synthases (LACS) to form acyl-CoAs, and these C16 and C18 acyl-CoAs undergo aliphatic chain elongation under the action of fatty acid elongase (FAE) enzyme complexes comprising ketoacyl-CoA synthases (KCS), hydroxyacyl-CoA dehydratases (HCD), ketoacyl-CoA reductases (KCR), and enoyl-CoA reductases (ECR), as well as ECERIFERUM2-LIKE (CER2-LIKE) proteins, to generate VLC acyl-CoAs [14,15,16,17,18,19,20]. VLC acyl-CoAs could either enter the alkane-forming pathway to generate VLC alkanes, secondary alcohols, and ketones or be converted to VLC primary alcohols by an alternative alcohol-forming pathway. VLC primary alcohols could be esterified with VLC acyl-CoAs to form wax esters [21,22,23,24,25,26]. These wax components, the VLC alcohols, alkanes, aldehydes, ketones, and esters, are then transported through the Golgi and trans-Golgi network (TGN)-trafficking pathways to the plasma membrane (PM), and then deposited to the cuticle via (PM) localized ATP-binding cassette transporter subfamily G (ABCG) subfamily half transporters and the lipid transfer proteins (LTPs) [27,28,29,30]. 

Accumulating evidence suggests that cuticular wax biosynthesis is tightly regulated at transcriptional levels, which has been overviewed recently [31]. The SHINE (SHN) clade of the AP2 domain transcription factors AtSHN1, AtSHN2, and AtSHN3 were first identified as transcriptional activators of cuticle lipid biosynthesis in the dicot model plant *Arabidopsis thaliana* [32,33,34]. Orthologs of AtSHN1, AtSHN2, and AtSHN3 have been characterized as key regulators of wax biosynthesis in other plant species such as *Hordeum vulgare*, *T. aestivum*, and *Physcomitrium patens* [35,36,37,38,39]. In addition, myeloblastosis (MYB)-type transcription factors AtMYB16, AtMYB30, AtMYB41, AtMYB94, AtMYB96, and AtMYB106 are revealed to become widely involved in the transcriptional regulation of wax biosynthesis in *A. thaliana* [1,2,3,40,41,42]. For instance, leaf cuticular wax components and the expressions of wax biosynthesis genes were altered in the *Arabidopsis* MYB30 knockout mutant and overexpressing lines [43]. Similarly, the knockdown of *MdMYB30,* an apple (*Malus domestica*) ortholog of the *AtMYB30* gene, attenuated wax deposition in apples [44]. In addition, wax biosynthesis is demonstrated to be regulated by MYB transcription factors in bread wheat. For instance, wheat MYB transcription factor TaMYB96/TaEPBM1 is recently identified as a wheat ortholog of the AtMYB96 and demonstrated to activate wax biosynthesis genes and potentiate wheat wax biosynthesis [45,46]. 

In this study, three closely homologous MYB genes *TaMYB30-5A*, *TaMYB30-5B*, and *TaMYB30-5D* were demonstrated to positively regulate wheat wax biosynthesis. The silencing of all endogenous *TaMYB30* genes resulted in attenuated wax accumulation and potentiated cuticle permeability in bread wheat. Wheat ortholog genes of *KCS1* and *ECR* (*TaKCS1-4A*, *TaKCS1-4B*, *TaKCS1-4D*, *TaECR-3A*, *TaECR-3B*, and *TaECR-3D*) were found to be essential for the wheat wax biosynthesis. Knockdown expression levels of all endogenous *TaKCS1* or *TaECR* genes led to reduced wax accumulation and enhanced cuticle permeability in bread wheat. In addition, TaMYB30 proteins were found to be transcriptional activators and could directly activate the expressions of wax biosynthesis genes *TaKCS1* and *TaECR*. These results strongly suggest that the transcriptional activator TaMYB30 positively regulates wheat wax biosynthesis, possibly via the transcriptional activation of wax biosynthesis genes *TaEDS1* and *TaSRAD1*.

## 2. Results

### 2.1. Identification of Wheat TaMYB30 Based on Homology with Arabidopsis AtMYB30

The MYB transcription factor AtMYB30 was demonstrated to be a key regulator of wax biosynthesis in the dicot model plant *Arabidopsis* [43]. In this study, we are interested in characterizing the function of the wheat homolog of AtMYB30 in the wax biosynthesis of bread wheat. To this end, we first searched the reference genome of the hexaploid bread wheat by using the amino acid sequence of *Arabidopsis AtMYB30* (At3g28910) as a query and obtained *TaMYB30*, the most closed homologs of *AtMYB30*, in bread wheat. Three highly homologous sequences of *TaMYB30* genes separately located on chromosomes 5A, 5B, and 5D were obtained from the genome sequence of the hexaploid wheat and designated as *TaMYB30-5A* (*TraesCS5A02G227400*), *TaMYB30-5B* (*TraesCS5B02G226100*), and *TaMYB30-5D* (*TraesCS5D02G234800*), respectively. As shown in Figure 1A, these predicted TaMYB30-5A, TaMYB30-5B, and TaMYB30-5D proteins shared about 45% identities with *Arabidopsis* AtMYB30. The TaMYB30-5A, TaMYB30-5B, and TaMYB30-5D proteins all contain two conserved MYB motifs at their N-termini (Figure 1B). The coding regions of these allelic *TaMYB30* genomic sequences all contained three exons and two introns (Figure 1C).

### 2.2. TaMYB30 Positively Regulates Wheat Wax Biosynthesis

To study the function of *TaMYB30* in wheat wax biosynthesis, we performed barley stripe mosaic virus (BSMV)-induced gene silencing (BSMV-VIGS) to silence all endogenous *TaMYB30* genes in the leaves of the wheat cultivar Yannong 999. BSMV*-γ*-infected wheat plants were employed as the negative controls. As revealed by the quantitative reverse-transcription polymerase chain reaction (qRT-PCR) assay, the TaMYB30 expression level was significantly reduced in the BSMV-*TaMYB30as*-infected wheat leaves (Figure 2A). We then conducted the gas chromatography-mass spectrometry (GC-MS) assay to measure cuticular wax constituents released from BSMV-*γ* and BSMV-*TaMYB30as* plants. As shown in Figure 2B, the amount of total cuticular wax was reduced from 12.1 μg cm^−2^ in the wheat leaves infected with BSMV-γ to 2.29 μg cm^−2^ in the BSMV-*TaMYB30as*-infected wheat plants (Figure 2B). Further wax component analyses revealed that the most abundant wax constituent, C28-alcohol, decreased from 8.05 μg cm^−2^ in the BSMV-γ wheat leaves to 1.05 μg cm^−2^ in the BSMV-*TaMYB30as* plants (Figure 2C). Likewise, other VLC fatty acids, alcohols, aldehydes, alkanes, and esters all showed an obvious reduction in the BSMV-*TaMYB30as*-infected wheat plants compared with BSMV-γ control plants (Figure 2C). These results indicated that the silencing of *TaMYB30* significantly attenuated wax biosynthesis in wheat leaves and suggested that *TaMYB30* positively regulates wheat wax biosynthesis.

Changes in wax deposition usually resulted in altered cuticle permeability. Compared with the BSMV-γ controls, the leaves of the BSMV-*TaMYB30as*-infected wheat plants displayed higher water loss rates with a loss of approximately 70% of the total water after 12 h (Figure 2D). Consistently, the chlorophyll leaching assay demonstrated that the BSMV-*TaMYB30as*-infected wheat plants released chlorophyll faster than the BSMV-γ-infected wheat plants (Figure 2E). Collectively, these results suggested that the silencing of *TaMYB30* resulted in attenuated wax biosynthesis and enhanced surface barrier properties in wheat leaves.

### 2.3. Homology-Based Identification of TaKCS1 and TaECR in Bread Wheat

Previous studies revealed that the *Arabidopsis* MYB transcription factor AtMYB30 could regulate the expressions of wax biosynthesis genes *AtKCS1* and *AtECR* [43]. We were interested in testing the potential regulation of TaMYB30 on these wax biosynthesis genes in bread wheat. To this end, we searched the reference genome of the hexaploid bread wheat by using the amino acid sequence of *Arabidopsis* AtKCS1 (At1g01120) and AtECR (At3g55360) as a query and obtained TaKCS1 and TaECR1, the most closed homologs of AtKCS1 and AtECR1, in bread wheat. Three highly homologous sequences of *TaKCS1* genes separately located on chromosomes 4A, 4B, and 4D were obtained from the genome sequence of the hexaploid wheat and designated as *TaKCS1-4A* (*TraesCS4A02G068400*), *TaKCS1-4B* (*TraesCS4B02G225500*), and *TaKCS1-4D* (*TraesCS4D02G226100*), respectively. Similarly, three highly homologous sequences of *TaECR* genes separately located on chromosomes 3A, 3B, and 3D were obtained from the genome sequence of the hexaploid wheat and designated as TaECR-3A, TaECR-3B, and TaECR-3D, respectively [33].

As shown in Figure 3A, these predicted TaKCS1-4A, TaKCS1-4B, and TaKCS1-4D proteins shared about 43% identities with *Arabidopsis* AtKCS1. In addition, the TaKCS1-4A, TaKCS1-4B, and TaKCS1-4D proteins all contain a central FAE1/Type III polyketide synthase-like protein (FAE1_CUT1_RppA) domain, and the C-terminal 3-Oxoacyl-[acyl-carrier-protein (ACP)] synthase III C terminal (ACP_syn_III_C) domain (Figure 3B). The coding regions of these allelic *TaKCS1* genomic sequences all contained one exon and zero introns (Figure 3C). The predicted TaECR-3A, TaECR-3B, and TaECR-3D proteins shared about 81% identities with *Arabidopsis* AtECR (Figure 3D). In addition, the TaECR-3A, TaECR-3B, and TaECR-3D proteins all contain a C-terminal 3-oxo-5-alpha-steroid 4-dehydrogenase (3-oxo-5_α-steroid 4-DH_C) domain (Figure 3E). The coding regions of these allelic *TaECR* genomic sequences all contained four exons and three introns (Figure 3F).

### 2.4. TaKCS1 and TaECR Positively Contribute to Wheat Wax Biosynthesis

To study the function of *TaKCS1 and TaECR* in wheat wax biosynthesis, we employed the BSMV-VIGS to silence all endogenous *TaKCS1* or *TaECR* genes in the leaves of the wheat cultivar Yannong 999. As shown in Figure 4A, the expression levels of *TaKCS1* or *TaECR* genes are significantly reduced in the BSMV-*TaKCS1as*- or BSMV-*TaECRas*-infected wheat plants (Figure 4A). We then measured cuticular wax constituents released from BSMV-*γ*, BSMV-*TaKCS1as*, or BSMV-*TaECRas* plants by GC-MS assay. As shown in Figure 4B, the amount of total cuticular wax decreased from 11.68 μg cm^−2^ in the BSMV-γ leaves to 2.89 μg cm^−2^ and 2.51 μg cm^−2^ in the wheat leaves infected with BSMV-*TaKCS1as* and BSMV-*TaECRas*, respectively (Figure 4B). Wax component analyses revealed that the amounts of VLC fatty acids, alcohols, aldehydes, alkanes, and esters were reduced significantly in the BSMV-*TaKCS1as* and BSMV-*TaECRas* leaves compared with the BSMV-γ controls (Figure 4C). These results collectively showed that the knockdown of *TaKCS1* and *TaECR* expression levels remarkably attenuated wax biosynthesis in wheat leaves and confirmed that *TaKCS1* and *TaECR* positively contribute to wheat wax biosynthesis.

Thereafter, we analyzed the cuticle permeability in the *TaKCS1*- or *TaECR*-silenced wheat plants by measuring the water loss rate and chlorophyll extracted. As shown in Figure 4D, the leaves of the BSMV-*TaKCS1as*- or BSMV-*TaECRas*-infected wheat plants displayed higher water loss rates than the BSMV-γ controls. Consistent with this, the chlorophyll leaching assay showed that the wheat leaves with silenced *TaKCS1* or *TaECR* genes released chlorophyll faster than the BSMV-γ controls (Figure 4E). Overall, these results suggested that the knockdown of the *TaKCS1* and *TaECR* expression levels led to compromised attenuated wax biosynthesis and enhanced surface barrier properties in wheat leaves.

### 2.5. TaMYB30 Activates Expression of TaKCS1 and TaECR

To determine the potential regulation of *TaMYB30* on the expression of *TaKCS1* and *TaECR* in bread wheat, we employed BSMV-VIGS to silence all endogenous *TaMYB30* genes in the leaves of wheat cultivar Yannong 999. Thereafter, expression levels of *TaKCS1* or *TaECR* genes were analyzed by qRT-PCR assay. As shown in Figure 5A, the silencing of *TaMYB30* genes resulted in above 70% decreases in the expression levels of *TaKCS1* and *TaECR*, suggesting that the MYB transcription factor TaMYB30 positively regulates the expressions of *TaKCS1 and TaECR.*

To quantify the transcriptional activity of TaMYB30 proteins, we performed the *Arabidopsis* leaf protoplast transfection assay, in which a luciferase (LUC) reporter was co-transfected with effector constructs expressing TaMYB30-5A, TaMYB30-5B, or TaMYB30-5D protein, and a control plasmid, to analyze the reporter luciferase activity (LucA). The Gal4 DNA-binding domain (DBD) was used to determine the basal LUC activity. As shown in Figure 5B, the LucA ratio has increased from 1 for the DBD control to above 2.2 under the expression of DBD-TaMYB30-5A, DBD-TaMYB30-5B, or DBD-TaMYB30-5D, indicating that TaMYB30 proteins exhibit transcriptional activator activity. 

It has been demonstrated that wheat MYB transcription factor TaMYB96/TaEPBM1, a homolog of TaMYB30, could directly bind to the cis-elements *MBS* and *Motif 1* [45,46]. Since *MBS* and *Motif 1* cis-elements were found in the promoter regions of *TaKCS1* and *TaECR* genes, we examined whether TaMYB30 proteins could directly bind to the *TaKCS1* and *TaECR* promoters and activate their expressions. In the *Arabidopsis* leaf protoplast transfection assay, LUC reporters containing promoter regions of *TaKCS1-4A*, *TaKCS1-4B*, *TaKCS1-4D*, *TaECR-3A*, *TaECR-3B*, or *TaECR-3D* genes were co-transfected with effector constructs overexpressing TaMYB30-5A, TaMYB30-5B, or TaMYB30-5D proteins, and a control plasmid. Wild-type promoter regions of *TaKCS1-4A*, *TaKCS1-4B*, *TaKCS1-4D*, *TaECR-3A*, *TaECR-3B*, or *TaECR-3D* genes harboring the cis-elements *MBS* and *Motif 1*, and *TaKCS1-4A*, *TaKCS1-4B*, *TaKCS1-4D*, *TaECR-3A*, *TaECR-3B*, or *TaECR-3D* promoter mutants containing mutated *MBS* and *Motif 1* cis-elements, were employed. As shown in Figure 5C,D, the LucA ratio obtained from LUC reporters containing wild-type *TaKCS1* and *TaECR* promoters increased to above 2.07 in the presence of over-accumulated TaMYB30 proteins, compared with 1 for the Gal4 DNA-binding domain (DBD) proteins and with less than 0.3 for the empty vector (EV) control. In contrast, the LucA ratio obtained from mutant *TaKCS1* and *TaECR* promoters was not significantly affected by over-accumulated TaMYB30 proteins (Figure 5C,D). This result implies that the transcriptional activators TaMYB30-5A, TaMYB30-5B, and TaMYB30-5D could directly bind to the promoter regions of *TaKCS1* and *TaECR* genes by recognizing the *MBS* and *Motif 1* cis-elements, and activate their expression in plant cells.

## 3. Discussion

### 3.1. TaMYB30 Is a Positive Regulator of Wax Biosynthesis in Bread Wheat

In this study, TaMYB30 is characterized as a positive regulator of wheat wax biosynthesis. The knockdown of *TaMYB30* expression using BSMV-VIGS resulted in attenuated wax biosynthesis and enhanced cuticle permeability. Increasing evidence revealed that wheat wax biosynthesis is governed by transcription factors and their interactors. For instance, the knockout or knockdown expression of the transcription factor gene *TaSHN1/WIN1* attenuated wheat wax biosynthesis, whereas the transgenic overexpression of the *TaSHN1/WIN1* gene resulted in enhanced wax accumulation in bread wheat [38,39,47]. The mediator component CYCLIN-DEPENDENT KINASE8 (TaCDK8) was demonstrated to interact and phosphorylate the transcription factor TaSHN1/WIN1 [38,48,49,50]. The silencing of *TaCDK8* by BSMV-VIGS assay resulted in attenuated wax accumulation in wheat leaves, indicating that TaCDK8 positively regulates wheat wax biosynthesis [38]. Interestingly, AtCDK8 interacts with AtSHN1 to potentiate wax biosynthesis in the dicot model plant *A. thaliana*, implying that the SHN1-CDK8 interacting module regulates wax biosynthesis in dicot and monocot plants [51]. The wheat R2R3-type MYB transcription factor TaMYB96/TaEPBM1 was revealed to trigger cuticular wax biosynthesis [45,46]. The TaGCN5-TaADA2 histone acetyltransferase module was shown to associate with TaMYB96/TaEPBM1 and mediate histone acetylation at the promoter regions of TaMYB96/TaEPBM1 target genes [45]. The knockdown of *TaGCN5* or *TaADA2* expression levels led to reduced wax accumulation, indicating that the TaGCN5-TaADA2 histone acetyltransferase module interacts with TaMYB96/TaEPBM1 to trigger wax biosynthesis [45]. Similarly, the CHD3-type chromatin remodeling factor TaCHR729 could interact with the basic helix-loop-helix (bHLH) transcription factor TaKPAB1 and mediate the deposition of the permissive epigenetic mark H3K4me3 at TaKCS6 promoters [52,53]. These studies demonstrated that transcription factors interact with other regulators such as mediator components and epigenetic components to fine-tune wheat wax biosynthesis. Therefore, it is intriguing to identify TaMYB30-interacting partners and characterize their potential roles in the regulation of wheat wax biosynthesis in future research.

### 3.2. Wheat TaKCS1 and TaECR Are Essential Components of Wax Biosynthetic Machinery

*TaKCS1* is identified as an important component of wax biosynthetic machinery in bread wheat. In *A. thaliana*, *AtKCS1* encodes a 3-ketoacyl-CoA synthase involved in wax biosynthesis [54]. The T-DNA-tagged *kcs1-1* mutant displayed a decrease of up to 80% in the levels of VLC wax alcohols and aldehydes, but much smaller effects on the major *Arabidopsis* wax components the C29 alkanes and on the total wax load, suggesting the potential redundancy in the KCS activities involved in *Arabidopsis* wax synthesis [54]. In contrast, the barley *eceriferum-zh* (*cer-zh*) mutant displays an obvious glossy and wax-deficient phenotype [55]. *CER-ZH* encodes the β-ketoacyl-CoA synthase HvKCS1, a barley ortholog of AtKCS1. In this study, the silencing of *TaKCS1* by BSMV-VIGS led to compromised wax biosynthesis, increased water loss rates, and enhanced chlorophyll leaching [55]. These results indicate the potential functional conservation of *KCS1* in monocot and dicot plants, and that the distinct contribution of KCS1 to total wax load in *Arabidopsis*, barley, and wheat might result from different wax compositions among these plant species. 

As an obligate biotrophic fungal pathogen, *Blumeria graminis forma specialis tritici* (*B.g. tritici*) has evolved sophisticated mechanisms to infect the host wheat plants. Increasing evidence suggests that wax signals from wheat epidermis could be harnessed by the host-adapted *B.g. tritici* fungus to promote pre-penetration infection processes [38,45,52]. For instance, the silencing of wheat wax biosynthesis genes *TaWIN1*, *TaKCS6* and *TaECR* resulted in the attenuated conidial germination of *B.g. tritici*. In this study, the knockdown of *TaMYB30* or *TaKCS1* expressions led to reduced cuticular wax biosynthesis. It is therefore intriguing to examine whether and how *TaMYB30* and *TaKCS1* affect pre-penetration events such as conidial germination in the infection of *B.g. tritici* in future studies.

The *Enoyl-CoA Reductase* (*ECR*) gene encodes the enoyl-CoA reductase that catalyzes the biosynthesis of VLC acyl-CoAs in the fatty acid elongase complex [56]. Recent phylogenetic analysis showed that *ECR* genes are highly conserved among land plants. The knockout of *ECR* genes could significantly attenuate wax biosynthesis in *Arabidopsis*, whereas the ectopic overexpression of *CsECR* cloned from the “Newhall” navel orange (*Citrus sinensis* [L.] Osbeck cv. Newhall) increased cuticular wax accumulation in the tomato (*Solanum lycopersicum*) and enhanced its drought resistance [57,58]. In this and previous studies, the silencing of *TaECR* by BSMV-VIGS resulted in reduced wax biosynthesis and enhanced cuticle permeability, as well as the attenuated germination of *B.g. tritici* conidia [45]. These data suggested that *ECR* plays a key role in the wax biosynthesis among dicot and monocot plants, and has potential values in crop breeding for improving stress resilience and disease resistance. 

### 3.3. TaMYB30 Activates Expression of TaKCS1 and TaECR1 to Positively Regulate Wheat Wax

Three closely homologous TaMYB30-5A, TaMYB30-5B, and TaMYB30-5D proteins were identified as transcriptional activators in this study. Importantly, TaMYB30-5A, TaMYB30-5B, or TaMYB30-5D could directly bind to the promoter regions of *TaKCS1* and *TaECR* genes by recognizing the *MBS* and *Motif 1* cis-elements, and stimulate their expression in plant cells. These data collectively suggested that TaMYB30 positively regulates wheat wax biosynthesis presumably via the transcriptional activation of *TaKCS1* and *TaECR* (Figure 5E). It was previously reported that the expressions of *Arabidopsis* wax biosynthesis genes such as *AtKCS1*, *AtECR*, etc. were reduced in the *AtMYB30*-knockout mutant but enhanced in the *AtMYB30*-overexpressing lines, indicating that the up-regulation of wax biosynthesis genes by MYB30 and its orthologs is conserved among dicots and monocots [43]. 

It was previously demonstrated that another R2R3-type MYB transcription factor TaMYB96/TaEPBM1 could activate the expression of *TaECR* genes and positively regulate wheat wax biosynthesis [45]. In this study, the transcriptional activator TaMYB30 was shown to directly recognize the promoter regions of *TaECR-3A*, *TaECR-3B*, and *TaECR-3D* genes, and to activate their expression, suggesting that a single wax biosynthesis gene might be targeted by different MYB transcription factors. Accumulating studies revealed that plant wax biosynthesis is tightly regulated at the transcriptional level and affected by developmental stages and environmental conditions. It is intriguing to examine the interplays among different MYB transcription factors in the regulation of wax biosynthesis in response to developmental and environmental cues in future research.

## 4. Materials and Methods

### 4.1. Plant Materials

The plant seedlings of bread wheat cultivar Yannong 999 were grown in a growth chamber under a 16-h/8-h, 20 °C/18 °C day/night cycle with 70% relative humidity. *A. thaliana* Col-0 used in this study was grown in the greenhouse under a 16-h/8-h light period at 23 ± 1 °C with 70% relative humidity.

### 4.2. Quantitative Real-Time PCR (qRT-PCR)

The cuticular wax composition analysis was performed as described [38]. Briefly, total RNA was extracted from the wheat leaves using the EasyPure Plant RNA kit (Transgenbiotech, Beijing, China) and was used to synthesize the cDNA template using the TransScript one-step gDNA removal and cDNA synthesis supermix (Transgenbiotech, Beijing, China). The Real-time PCR assay was performed using the ABI real-time PCR system with the qPCR Master Mix (Invitrogen, Carlsbad, CA, USA). The expression of *TaGADPH* was set as the internal control and the expression levels of *TaGAPDH*, *TaMYB30*, *TaKCS1*, and *TaECR* were analyzed using the primer pairs 5′-TTAGACTTGCGAAGCCAGCA-3′/5′-AAATGCCCTTGAGGTTTCCC-3′, 5′-TGCTCGTCTCCTACATCCA-3′/5′-TATCGCCGCCCAACGGTTG-3′, 5′-CTACTCCTTCGTCCGCCTC-3′/5′-GTGATCTTGGTCTGGAACG-3′, and 5′-TGAAGGTCTCCGTCGTGTCC-3′/5′-CAGAAGAAGAGCGTGCTGTAG-3′, respectively.

### 4.3. BSMV-Mediated Gene Silencing 

Antisense fragments of *TaMYB30*, *TaKCS1*, and *TaECR* were cloned into the pCa-γbLIC vector to create the BSMV-Ta*MYB30*as, BSMV-Ta*KCS1*as, and BSMV-*TaECR*as constructs using the primer pairs 5′-AAGGAAGTTTATAGAAGAACTCACTGGGGTC-3′/5′-AACCACCACCACCGTGCCAGCAGCCGCAGTGCTC-3′, 5′-AAGGAAGTTTATGTG CACGCAGAAGTGCTCG-3′/5′-AACCACCACCACCGTGGGACGAGTGCTTCAAGT GC-3′, and 5′-AAGGAAGTTTGGAGGAAGAATCACCCATCG-3′/5′-AACCACCACC ACCGT TCAACATTGTGACATGCGCAAAC-3′, respectively. The BSMV-mediated gene silencing in wheat leaves was performed as described by Kong and Chang [38].

### 4.4. Water Loss Rate and Chlorophyll Leaching Tests

The water loss rate tests were performed as described previously [59,60]. Briefly, the leaves (*n* = 5) with BSMV virus symptoms about 2 weeks post-BSMV-infection were dipped in ultrapure water under dark conditions for 1 h to maintain the stomatal closure, and leaf weights were then measured per 1 h for 12 h. The water loss rate tests were performed as described [59,61]. Wheat plants (*n* = 5) with BSMV virus symptoms about 2 weeks post-BSMV-infection were kept in the dark and immersed in ultrapure water for 1 h. Then leaves were detached and dipped in 80% ethanol, and the amount of chlorophyll extracted was measured at 647 and 664 nm with a spectrophotometer (Cary 60 UV-Vis, Agilent Technologies, Santa Clara, CA, USA) every 60 min for 12 h.

### 4.5. Wax Composition Analysis

The cuticular wax composition analysis was performed as described [38]. The leaves (*n* = 5) with BSMV virus symptoms about 2 weeks post-BSMV-infection were dipped into chloroform (Merck, Rahway, NJ, USA). The extracts dried under N_2_ were derivatized at 70 °C for 30 min through reaction with bis-N,O- trimethylsilyl trifluoroacetamide and analyzed with a capillary GC (5890 Series II, Agilent Technologies) and a flame ionization detector (6890 N, Agilent Technologies) with a mass spectrometer (MSD 5973, Agilent Technologies) as previously described [38]. Quantification was based on FID peak areas relative to the internal standard n-Tetracosane (Merck) and the dry, delipidated tissue weights.

### 4.6. Arabidopsis Protoplast Transactivation Assay

The isolation and transformation of the *Arabidopsis* protoplast were conducted as described previously by Zhi et al. [62]. The vector pRT-DBD and pRT were employed to generate construct expression proteins with and without DBD fusion, respectively. The effector constructs, including pRT, pRT-BD, and derivatives, were mixed with the reporter gene and co-delivered into the *Arabidopsis* protoplasts by transfection. At 48 h after transfection, the LUC activity was analyzed using a Promega dual-luciferase reporter assay system (Promega, Madison, WI, USA, E1910) following the manufacturer’s manual.

## Figures and Tables

**Figure 1 ijms-24-10235-f001:**
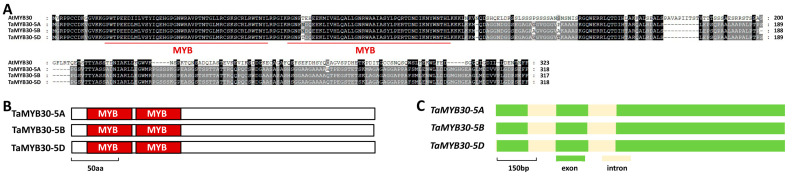
Homology-based identification of TaMYB30 in bread wheat. (**A**) Protein sequence comparison of wheat TaMYB30 and *Arabidopsis* AtMYB30. (**B**) Domain structure of wheat TaMYB30 proteins. (**C**) Gene architectures of wheat *TaMYB30* genes.

**Figure 2 ijms-24-10235-f002:**
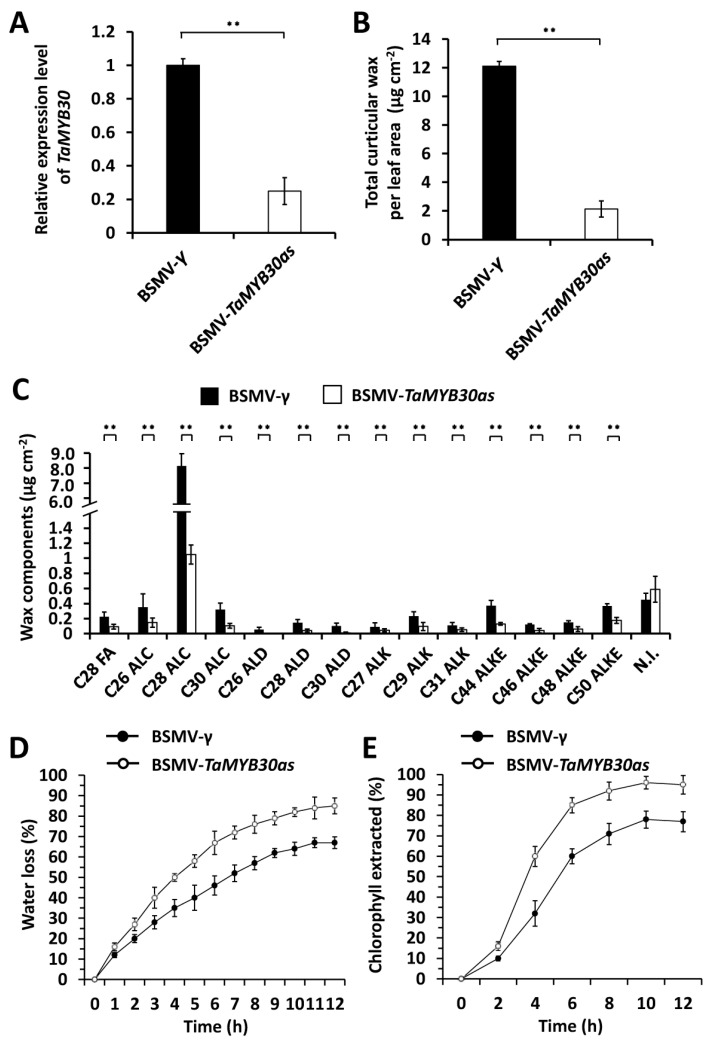
Functional analyses of wheat *TaMYB30* in wheat wax biosynthesis. (**A**) qRT-PCR analysis of *TaMYB30* expression levels in the wheat leaves infected with BSMV-*γ* or BSMV-*TaMYB30as*. (**B**) Total cuticular wax amounts in the wheat leaves infected with BSMV-γ or BSMV-*TaMYB30as*. (**C**) Amounts of major wax components in the BSMV-*γ* and BSMV-*TaMYB30as* wheat leaves. FA, fatty acid; ALC, alcohol; ALD, aldehyde; ALK, alkane; ALKE, alkyl ester; N. I., not identified compound. (**D**) Water loss rates and (**E**) chlorophyll extraction levels were measured using leaves infected with BSMV-γ or BSMV-Ta*MYB30as*. For (**A**–**E**), three independent biological replicates were statistically analyzed for each treatment (*t*-test; ** *p* < 0.01).

**Figure 3 ijms-24-10235-f003:**
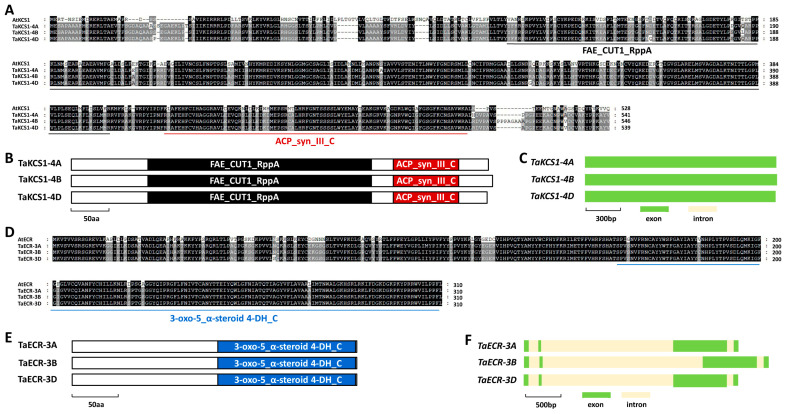
Identification of wheat TaKCS1 and TaECR based on homology with *Arabidopsis* AtKCS1 and AtECR. (**A**) Protein sequence comparison of wheat TaKCS1 and *Arabidopsis* AtKCS1. (**B**) Domain structure of wheat TaKCS1 proteins. (**C**) Gene architectures of wheat *TaKCS1* genes. (**D**) Protein sequence comparison of wheat TaECR and *Arabidopsis* AtECR. (**E**) Domain structure of wheat TaECR proteins. (**F**) Gene architectures of wheat *TaECR* genes.

**Figure 4 ijms-24-10235-f004:**
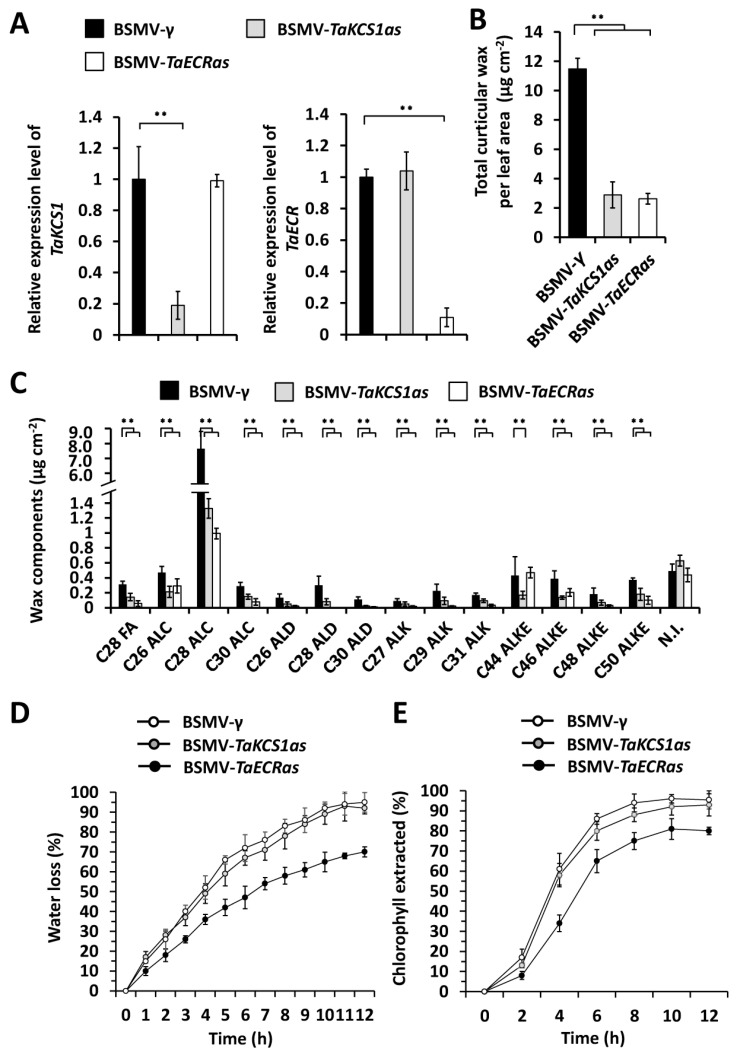
Functional analyses of wheat *TaKCS1* and *TaECR* in wheat wax biosynthesis. (**A**) qRT-PCR analysis of *TaKCS1* and *TaECR* expression levels in the wheat leaves infected with BSMV-*γ*, BSMV-*TaKCS1as*, or BSMV-*TaECRas*. (**B**) Total cuticular wax amounts in the wheat leaves infected with BSMV-*γ*, BSMV-*TaKCS1as*, or BSMV-*TaECRas*. (**C**) Amounts of major wax components in the BSMV-*γ*, BSMV-*TaKCS1as*, and BSMV-*TaECRas* wheat leaves. FA, fatty acid; ALC, alcohol; ALD, aldehyde; ALK, alkane; ALKE, alkyl ester; N. I., not identified compound. (**D**) Water loss rates and (**E**) chlorophyll extraction levels were measured using leaves infected with BSMV-*γ*, BSMV-*TaKCS1as*, or BSMV-*TaECRas.* For (**A**–**E**), three independent biological replicates were statistically analyzed for each treatment (*t*-test; ** *p* < 0.01).

**Figure 5 ijms-24-10235-f005:**
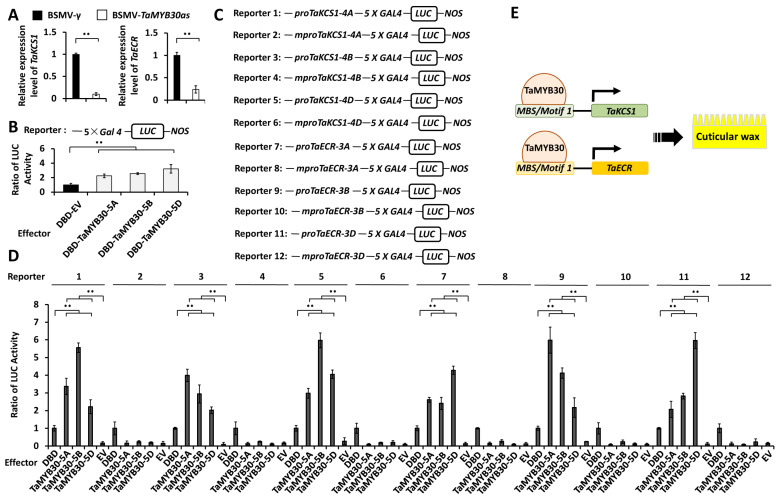
Analysis of the transcriptional activation of *TaKCS1* and *TaECR* genes by TaMYB30. (**A**) qRT-PCR analysis of *TaKCS1* and *TaECR* expression levels in *TaMYB30*-silenced wheat leaves. (**B**) Transactivation activity analysis of TaMYB30 in *Arabidopsis* cells. The LUC reporter used in this assay contains 5× Gal4 UAS. Indicated effectors were co-expressed with the LUC reporter in the *Arabidopsis* protoplast by transfection. The LUC activity was then examined using a dual-luciferase reporter assay system. (**C**) Schematic depiction of the LUCIFERASE (LUC) reporter containing a wild-type or mutant (labeled as pro and mpro) promoter fragments of *TaKCS1* and *TaECR* genes. (**D**) Transactivation of *TaKCS1* and *TaECR* by TaMYB30 in *Arabidopsis* protoplast. LUC activity was normalized to that obtained from protoplasts expressing DBD alone. For (**A**,**B**,**D**), three independent biological replicates were statistically analyzed for each treatment (*t*-test; ** *p* < 0.01). (**E**) A proposed model of the action of TaMYB30 in the regulation of wheat wax biosynthesis. Transcription activator TaMYB30 directly binds to the promoter regions of *TaKCS1* and *TaECR* genes by recognizing the *MBS* and *Motif 1* cis-elements, and activates their expressions, which leads to the potentiated wax biosynthesis.

## Data Availability

Data presented here are available on request from correspondence.

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
