# Peer review of "Transcription Factor TaMYB30 Activates Wheat Wax Biosynthesis"

_ijms, 2023, doi:10.3390/ijms241210235_

Round 1
Reviewer 1 Report
The manuscript is interesting and in the scope of the journal, it contains valuable results. The mechanisms of regulation of cuticular wax biosynthesis have not been completely elucidated do far, therefore, every attempt for such studies provides new valuable data. The methods are well described, the results are properly presented and the conclusions are convincing.
Line 97, 129, 138, 172, 236. The caption of the Figures should not be written in bold (besides Figure and its number).
Line 331. Please indicate what type of Arabidopsis thaliana was used in this study (so many various forms and mutants of this model plant exists, that it is necessary to precise that).
Author Response
Review 1 # Comments and Suggestions for Authors
The manuscript is interesting and in the scope of the journal, it contains valuable results. The mechanisms of regulation of cuticular wax biosynthesis have not been completely elucidated do far, therefore, every attempt for such studies provides new valuable data. The methods are well described, the results are properly presented and the conclusions are convincing.
- Response: Thank you very much for these encouraging comments. We have made extensive revision of this manuscript according to referees’ comments. Hopefully, this version could meet the standard for publication.
Line 97, 129, 138, 172, 236. The caption of the Figures should not be written in bold (besides Figure and its number).
- Response: Many thanks. The revised caption of the Figures has been written in thin.
Line 331. Please indicate what type of Arabidopsis thaliana was used in this study (so many various forms and mutants of this model plant exists, that it is necessary to precise that).
- Response: We fully agree with the Reviewer. Ecotype (Col–0) of Arabidopsis thaliana used in this study has been indicated in the revised manuscript.
Reviewer 2 Report
Dear Authors,
Reviewer comments ijms-2460438
The manuscript entitled „Transcriptional activator TaMYB30 positively regulates wheat wax biosynthesis“ represents a valuable study providing direct experimental evidence on a positive effect of TaMYB30 transcription factor on expression of KCS1 and ECR genes encoding enzymes involved in very-long chain fatty acid (VLCFA) biosynthesis which form waxes. For functional studies of TaMYB30, TaKCS1 and TaECR genes, BSMV-mediated gene sielncing approach to create BSMV-TaMYB30as, BSMV-TaKCS1as, and BSMV-TaECRas constructs were used. A comparison of BSMV-γ transformed plants with BSMV-TaMYBS30as revealed a significant negative effect of BSMV-mediated TaMYB30 gene silencing on total cuticular wax and the levels of the individual wax components.
I can recommend the present manuscript for publication in International Journal of Molecular Sciences since I have no major comments on the present mansucript.
I have only a few minor comments on the present manuscript which are given below:
1/ Plant materiál: In Materials and methods section, part 4.1. Plant materials, the source of bread wheat cultivar Yannong999, i.e., the institution from which it was obtained has to be specified.
2/ I think that some model scheme proposing the impacts of TamYB30 on the expression of KCS1 and ECR involved in very-long chain (VLC) fatty acid biosynthesis and wax biosynthesis and accumualtion should be provided as a figure or a graphical abstract.
3/ Formal comments on the text related to English language and style:
Introduction, line 37: Modify the word form „firstly“ to „first“ in the statement „“Briefly, C16 and C18 fatty acids trafficked from the plastid are first esterified with coenzyme A (CoA)…“
Line 155: Add the word „respectively“ at the end of the statement „Three highly homologous sequences of TaKCS1 genes separately located on chromosomes 4A, 4B, and 4D were obtained from the genome sequence of the hexaploid wheat and designated as TaKCS1-4A…., TaKCS1-4B…., nad TaKCS1-4D…., repectively.“
Line 157: Add the word „respectively“ at the end of the statement „Simialrly, three highly homologous sequences of TaECR genes separately located on chromosomes 3A, 3B, and 3D were obtained from the genome sequence of the hexaploid wheat and designated as TaECR-3A, TaECR-3B, and TaECR-3D, respectively [33].“
Materials and methods, line 341: Correct the abbreviation used for the housekeeping gene glucose phosphate dehydrogenase „TaGAPDH“ (not „TaGADPH“).
Materials and methods, line 346: Add the word „respectively“ at the end of the statement „The expression of TaGAPDH was set as the internal control and expression levels of TaGAPDH was set as the internal control and expression levels of TaGAPDH, TaMYB30, TaKCS1, and TaECR were analyzed using the primer pairs…….., respectively.“
Materials and methods, line 370: Write „30 min“, not „30 mins“.
Materials and methods, line 383: Add a comma between the words „DBD fusion“ and „respectively.“
Final recommendation: Accept after a minor revision.

Dear Authors,
Reviewer comments ijms-2460438
The manuscript entitled „Transcriptional activator TaMYB30 positively regulates wheat wax biosynthesis“ represents a valuable study providing direct experimental evidence on a positive effect of TaMYB30 transcription factor on expression of KCS1 and ECR genes encoding enzymes involved in very-long chain fatty acid (VLCFA) biosynthesis which form waxes. For functional studies of TaMYB30, TaKCS1 and TaECR genes, BSMV-mediated gene sielncing approach to create BSMV-TaMYB30as, BSMV-TaKCS1as, and BSMV-TaECRas constructs were used. A comparison of BSMV-γ transformed plants with BSMV-TaMYBS30as revealed a significant negative effect of BSMV-mediated TaMYB30 gene silencing on total cuticular wax and the levels of the individual wax components.
I can recommend the present manuscript for publication in International Journal of Molecular Sciences since I have no major comments on the present mansucript.
I have only a few minor comments on the present manuscript which are given below:
1/ Plant materiál: In Materials and methods section, part 4.1. Plant materials, the source of bread wheat cultivar Yannong999, i.e., the institution from which it was obtained has to be specified.
2/ I think that some model scheme proposing the impacts of TamYB30 on the expression of KCS1 and ECR involved in very-long chain (VLC) fatty acid biosynthesis and wax biosynthesis and accumualtion should be provided as a figure or a graphical abstract.
3/ Formal comments on the text related to English language and style:
Introduction, line 37: Modify the word form „firstly“ to „first“ in the statement „“Briefly, C16 and C18 fatty acids trafficked from the plastid are first esterified with coenzyme A (CoA)…“
Line 155: Add the word „respectively“ at the end of the statement „Three highly homologous sequences of TaKCS1 genes separately located on chromosomes 4A, 4B, and 4D were obtained from the genome sequence of the hexaploid wheat and designated as TaKCS1-4A…., TaKCS1-4B…., nad TaKCS1-4D…., repectively.“
Line 157: Add the word „respectively“ at the end of the statement „Simialrly, three highly homologous sequences of TaECR genes separately located on chromosomes 3A, 3B, and 3D were obtained from the genome sequence of the hexaploid wheat and designated as TaECR-3A, TaECR-3B, and TaECR-3D, respectively [33].“
Materials and methods, line 341: Correct the abbreviation used for the housekeeping gene glucose phosphate dehydrogenase „TaGAPDH“ (not „TaGADPH“).
Materials and methods, line 346: Add the word „respectively“ at the end of the statement „The expression of TaGAPDH was set as the internal control and expression levels of TaGAPDH was set as the internal control and expression levels of TaGAPDH, TaMYB30, TaKCS1, and TaECR were analyzed using the primer pairs…….., respectively.“
Materials and methods, line 370: Write „30 min“, not „30 mins“.
Materials and methods, line 383: Add a comma between the words „DBD fusion“ and „respectively.“
Final recommendation: Accept after a minor revision.
Author Response
Review 2 # Comments and Suggestions for Authors
The manuscript entitled ‘Transcriptional activator TaMYB30 positively regulates wheat wax biosynthesis’ represents a valuable study providing direct experimental evidence on a positive effect of TaMYB30 transcription factor on expression of KCS1 and ECR genes encoding enzymes involved in very-long chain fatty acid (VLCFA) biosynthesis which form waxes. For functional studies of TaMYB30, TaKCS1 and TaECR genes, BSMV-mediated gene sielncing approach to create BSMV-TaMYB30as, BSMV-TaKCS1as, and BSMV-TaECRas constructs were used. A comparison of BSMV-γ transformed plants with BSMV-TaMYBS30as revealed a significant negative effect of BSMV-mediated TaMYB30 gene silencing on total cuticular wax and the levels of the individual wax components. I can recommend the present manuscript for publication in International Journal of Molecular Sciences since I have no major comments on the present manuscript.
- Response: Thank you very much for these encouraging comments. We have made extensive revision of this manuscript according to referees’ comments. Hopefully, this version could meet the standard for publication.
I have only a few minor comments on the present manuscript which are given below:
1/ Plant materiál: In Materials and methods section, part 4.1. Plant materials, the source of bread wheat cultivar Yannong999, i.e., the institution from which it was obtained has to be specified.
- Response: Many thanks. The source of bread wheat cultivar Yannong999 has been indicated in the revised manuscript.
2/ I think that some model scheme proposing the impacts of TaMYB30 on the expression of KCS1 and ECR involved in very-long chain (VLC) fatty acid biosynthesis and wax biosynthesis and accumualtion should be provided as a figure or a graphical abstract.
- Response: We fully agree with the Reviewer. A model scheme proposing the action of TaMYB30 in the regulation of wheat wax biosynthesis has been included in the revised Figure 5.
3/ Formal comments on the text related to English language and style:
Introduction, line 37: Modify the word form ‘firstly’ to ‘first’ in the statement ‘Briefly, C16 and C18 fatty acids trafficked from the plastid are first esterified with coenzyme A (CoA)…’
- Response: Many thanks. We have modified the word form ‘firstly’ to ‘first’ in the revised statement.
Line 155: Add the word ‘respectively’ at the end of the statement ‘Three highly homologous sequences of TaKCS1 genes separately located on chromosomes 4A, 4B, and 4D were obtained from the genome sequence of the hexaploid wheat and designated as TaKCS1-4A…., TaKCS1-4B…., nad TaKCS1-4D…., repectively.’
- Response: Many thanks. The word ‘respectively’ has been added at the end of the revised statement.
Line 157: Add the word ‘respectively’at the end of the statement ‘Simialrly, three highly homologous sequences of TaECR genes separately located on chromosomes 3A, 3B, and 3D were obtained from the genome sequence of the hexaploid wheat and designated as TaECR-3A, TaECR-3B, and TaECR-3D, respectively [33].’
- Response: Yes, the word ‘respectively’ has been added at the end of the revised statement.
Materials and methods, line 341: Correct the abbreviation used for the housekeeping gene glucose phosphate dehydrogenase ‘TaGAPDH’(not ‘TaGADPH’).
- Response: Many thanks. We have corrected this mistake in the revision.
Materials and methods, line 346: Add the word ‘respectively’at the end of the statement ‘The expression of TaGAPDH was set as the internal control and expression levels of TaGAPDH was set as the internal control and expression levels of TaGAPDH, TaMYB30, TaKCS1, and TaECR were analyzed using the primer pairs…….., respectively.’
- Response: We thank the Reviewer for identifying this mistake. The word ‘respectively’ has been added at the end of the revised statement.
Materials and methods, line 370: Write ‘30 min’, not ‘30 mins’.
- Response: Thank you very much. We have corrected this mistake in the revision.
Materials and methods, line 383: Add a comma between the words ‘DBD fusion’and ‘respectively.’
- Response: Many thanks. A comma has been added between the words ‘DBD fusion’and ‘respectively.’ in the revised mansucript.